Conservation paleobiology on Minami-Daito Island, Okinawa, Japan: anthropogenic extinction of cave-dwelling bats on a tropical oceanic island

Kimura Yuri ykimura@kahaku.go.jp 1 2
Fukui Dai 3
Yoshiyuki Mizuko 4
Higashi Kazuaki 5
1 Department of Geology and Paleontology, National Museum of Nature and Science , Tsukuba , Ibaraki , Japan
2 Institut Català de Paleontologia Miquel Crusafont, ICTA-ICP , Barcelona , Spain
3 The University of Tokyo Hokkaido Forest, Graduate School of Agricultural and Life Sciences, The University of Tokyo , Furano , Hokkaido , Japan
4 Department of Agriculture, Tokyo University of Agriculture , Atsugi , Kanagawa , Japan
5 Office Key Point , Minami-Daito , Okinawa , Japan
Hedrick Brandon
Electronic publication date: 2022 Jan 27
Publication date: 2022
Volume: 10
Electronic Location ID: e12702
Received 2021 Sep 20; Accepted 2021 Dec 7
Copyright: ©2022 Kimura et al.
Copyright year: 2022
Copyright holder: Kimura et al.
License: This is an open access article distributed under the terms of the Creative Commons Attribution License, which permits unrestricted use, distribution, reproduction and adaptation in any medium and for any purpose provided that it is properly attributed. For attribution, the original author(s), title, publication source (PeerJ) and either DOI or URL of the article must be cited.
License URL: https://creativecommons.org/licenses/by/4.0/

Keywords: Anthropogenic extinction, Extirpation, Chiroptera, Conservation paleobiology, Stable carbon isotopes, FTIR, Fossil guano, Insular mammals, Endemic species, Oceanic island

Funding: I+D+i PID2020-117289GBI00 funded by MCIN/ AEI/10.13039/501100011033/ The Fujiwara Natural History Foundation (2018) The National Museum of Nature and Science JSPS KAKENHI 18K13650 The Generalitat de Catalunya (CERCA Programme) The 26th PRO NATURA FUND Asahi Glass Foundation JSPS KAKENHI 20H01979 This project was conducted as part of project I+D+i PID2020-117289GBI00 funded by MCIN/ AEI/10.13039/501100011033/. Yuri Kimura was financially supported by the Fujiwara Natural History Foundation (2018), the National Museum of Nature and Science as part of a research project called “Chemical Stratigraphy and Dating as a Clue for Understanding the History of the Earth and Life”, JSPS KAKENHI Grant Number 18K13650, and the Generalitat de Catalunya (CERCA Programme). Dai Fukui was financially supported by the 26th PRO NATURA FUND, Asahi Glass Foundation, and JSPS KAKENHI Grant Number 20H01979. The funders had no role in study design, data collection and analysis, decision to publish, or preparation of the manuscript.

==============================
Background

With strong environmental and geographic filtration, vertebrates incapable of flying and swimming are often extirpated from island ecosystems. Minami-Daito Island is an oceanic island in Okinawa, Japan that harbors the Daito flying fox (Pteropus dasymallus daitoensis), a subspecies of the fruit bat and the only extant mammal endemic to the island. However, the skeleton of a cave-dwelling bat Rhinolophus sp. and fossil guano were briefly reported in a previous study.

Methods

Here, we present evidence for the anthropogenic extirpation of two species of cave-dwelling bats (Miniopterus sp. & Rhinolophus sp.) from Minami-Daito Island. Our goal is to reliably constrain the ages of the extirpated bat species by a multiproxy approach. Because skeletal materials did not preserve sufficient bone collagen for direct radiocarbon dating, we alternatively examined guano-like deposits based on SEM observation and Fourier-transform infrared spectroscopy (FTIR) along with stable carbon and nitrogen isotope analyses for possible indirect dating. We also examined stable carbon isotopes in bone apatite, assuming that an isotopic signal of C4 plants on the bat bones links to sugarcane plantation on the island based on the historical knowledge that early human settlers quickly replaced the island’s native C3 forests with sugarcane (C4 perennial grass) plantation from 1900 onward.

Results

Our cave survey documents the remains of Miniopterus sp. from the island for the first time. Based on the unique taphonomic conditions (unpermineralized bones, disarticulated skeletons closely scattered without sediment cover, various degrees of calcite crystal growth around bones) and a radiocarbon age of a humic sample, we suggest that the maximum age constraint of Miniopterus sp. and Rhinolophus sp. is 4,640 calBP. Based on a series of analyses, we conclude that the guano-like deposits are composed not of bat guano but mainly of humic substances; however, a hydroxyapatite crust associated with bat-lying stalagmites may be derived from bat feces. Stable carbon isotope analysis of bone apatite revealed C4 signals in various degrees, confirming that small populations of cave-dwelling bats persisted on Minami-Daito Island after 1900.

Conclusions

The results of this study indicate that these populations remained rather small and did not leave many generations and that the estimated ages can be bracketed from 4,640 calBP to the post-1900 (perhaps, until the 1950s). They likely faced a continuously high mortality risk due to severe anthropogenic stresses on the island, where most of the forests were turned into sugarcane plantations within a few decades in the early 20th century. A result of hearing surveys to local residents suggests the latest remnants most likely disappeared on the island concurrently with the introduction of chemical pesticides after World War II.

Introduction

Insular systems such as remote islands, volcanic lakes, and deep caves, have offered natural testing grounds for understanding how migration, speciation, and extinction function in the long-term ecological dynamics of life (Van der Geer et al., 2010). Generally, vertebrate animals capable of flying, swimming, or more passively rafting in water compose native faunas in these systems, and geographic barriers and environmental filters yield species-poor communities with a high percentage of endemic taxa (e.g., Albert et al., 2017; Van der Geer et al., 2010). In these settings, human colonization has increased extinction risks of native species by economic activities and bringing introduced species in the natural ecosystem. Thus, long-term monitoring and quantification of species richness and biodiversity on these islands have gained recent attention regarding conservation against further anthropogenic extinction (e.g., Allentoft et al., 2014; Wood et al., 2017; Carpenter et al., 2020).

The Daito Islands in Japan are an archipelago of oceanic islands that have been separated from the closest landmass since their emergence. Historically, these islands were uninhabited by humans until the first exploitation over 120 years ago in 1899–1900. Prior to the arrival of humans, dense native forests comprising subtropical fan palm trees (Livistona chinensis var. amanoi), Neolitsea sericea var. argentea, banyans, and the evergreen shrub (Excoecaria formosana var. daitoinsularis) harbored endemic species of birds, insects, and fruit bats. By the 1930s, the rapid development of sugarcane plantation along with the filling of underground caves and sinkholes for land improvement and military construction caused severe and irreversible changes in vegetation (Editorial Committee of the History of Minami-daito Village, 1990). At present, any remaining natural forests are limited to the rims of the islands (Fig. 1A; Google Earth, 2020). Under a combination of high anthropogenic stresses and biogeographical constraints, the Daito flying fox (Pteropus dasymallus daitoensis), a subspecies of the Ryukyu flying fox (Pteropus dasymallus), is the only extant mammal species inhabiting the Daito Islands.

Figure 1 Map of Minami-Daito Island, route map of a privately-owned Cave A, cave map of Hoshino Cave.

(A) Satellite map (Google Earth, 2020); (B) Map of Ryukyu Islands; (C) Entrance of Cave A; (D) Route map of Cave A; (E) Map of Hoshino Cave (simplified from a map of Ehime University Expedition Club, 1972). Solid circles indicate sampling localities of guano-like deposits, and open stars indicate sampling localities of cave-dwelling bat bones.

Although the endangered Daito flying fox is the only native mammal on the Daito Islands today, Shimojana (1978) reported a skeleton of a cave-dwelling bat species found in Hoshino Cave, the only tourist cave on Minami-Daito Island, and classified the skeleton as Rhinolophus sp. Shimojana (1978) also noted that large amounts of fossil bat guano were present in the cave. Cave-dwelling bats, including fossils and extant ones, have not been documented on the island since then. Recently, based on information about additional skeletons of insectivorous bats in a different cave on the same island, we collected skeletal remains that belong to two bat species of different sizes.

In this study, we aim to constrain the ages of the extirpated bat species using a multiproxy approach. Unfortunately, bone collagen was not preserved in the collected bat bones (details in the Materials and Methods section), which makes direct radiocarbon dating impossible. Alternatively, for indirect radiocarbon dating, we examined (1) the guano-like deposits based on SEM observation and Fourier-transform infrared spectroscopy (FTIR) along with stable carbon and nitrogen isotope analyses to test the hypothesis that the “guano deposits” observed by Shimojana (1978) are truly fossil bat guano and (2) conduct stable carbon isotope analysis in bone apatite of the bats to detect a signal of C4 plants. Stable carbon isotopes in bone apatite reflect consumed diet of the bats and are not directly related to age determination. For age constraints of the extirpated bats, we take advantage of the historical fact that the island’s native C3 forests were quickly deforested and replaced with sugarcane (C4 perennial grass) plantation from the first development by early human settlers from 1900 onward (details in the Materials and Methods section). It is assumed here that a signal of C4 plants in the bat bones was caused by the presence of sugarcane plantation on the island and thus that the bat individuals were alive after 1900. We further discuss that the extirpation of the cave-dwelling bats was probably caused by high anthropogenic stresses on the small island.

Geological Setting

The Daito Islands are oceanic islands located on the Philippine Sea Plate, nearly 400 km southeast of Okinawa Island in Okinawa Prefecture, Japan, comprising Minami-Daito Island (Minamidait o ¯jima, 30.74 km2), Kita-Daito Island (Kitadait o ¯jima, 12.71 km2), and Oki-Daito Island (Okidait o ¯jima, 1.19 km2), listed in order of decreasing surface area (Figs. 1A–1B; Editorial Committee of the History of Minami-Daito Village, 1990). Oki-Daito Island is uninhabitable because of decades of firing exercises by the United States Navy, so ecological studies have only been conducted on Minami-Daito Island and Kita-Daito Island. The Daito Islands are atolls that drifted to its current position from near the equator where coral reefs began to accumulate over 50 million years ago (Klein & Kobayashi, 1980; Seno & Maruyama, 1984) and were uplifted during the Pliocene and Pleistocene epochs on the forebulge of the Philippine Sea Plate prior to subduction along the Ryukyu Trench (Ohde & Elderfield, 1992). Based on samples from boring surveys on Kita-Daito Island between 1934 to 1936, the carbonate deposits on the islands are up to 430 m thick, and the oldest strata are early Miocene in age (Ohde & Elderfield, 1992).

On Minami-Daito Island, Urushibara-Yoshino (2012) recognized two informal lithostratigraphic units of dolomitized limestone on the island, including the “Lower Daito Layer” and the more fossiliferous “Upper Daito Layer,” which are separated by an unconformity that was interpreted as the surface of karstification. Differential erosional rates of the dolomitized limestone layers have formed the unique topography of Minami-Daito Island that is characterized by basin-shaped lowlands (“Hagu Shita” in local vernacular) of the heavily karstified Lower Daito Layer surrounded by topographic highlands (“Hagu Ue” in local vernacular) of the Upper Daito Layer averaging 40m above sea level that form rims along the coast (Urushibara-Yoshino, 2012; Nambu et al., 2003). A coastal bench ∼10 m above sea level is overlain by a thin coral limestone dated to ∼125 ka, corresponding to the Marine Isotope Stage 5e during the last interglacial period in the Pleistocene (Ota et al., 1991), whereas the lowest coastal bench at ∼3 m above sea level may have formed more recently during a mid-Holocene sea-level highstand, the age of which is 6,500–5,000 BP (Urushibara-Yoshino, 2012).

Materials and Methods

Skeletal remains in the caves of Minami-Daito Island

There are countless sinkholes (dolines) and caves on Minami-Daito Island, some of which preserve the remains of extirpated bats. We collected vertebrate skeletal remains and bat guano-like deposits from two of these caves, which are a public cave called Hoshino Cave (also called Hoshinodo ¯ or Hoshinodo Cave, as “do ¯” means “cave” in Japanese) and a privately-owned cave informally called “Cave A” (Fig. 1A). Fieldwork was conducted in January 2016 and January 2018 because the CO2 concentration levels in the caves are lowest during the winter. All fieldwork was conducted outside national parks or restricted areas designated by the Nature Conservation Act.

Cave A is accessible via several spots where the ceiling has collapsed, but we entered through the main natural entrance, which is located in the Hagu Shita basin near the basin-side edge of the Hagu Ue rim (Fig. 1C). Because there is no published map for Cave A, we made a simple route map from the main entrance to our sample locations (Fig. 1D). Skeletal remains of insectivorous bats were collected from a small hall about 60 m north of the main entrance. Generally, skeletal remains are scattered, suggesting that these individuals were not transported by water flow (Figs. 2A–2B). In Cave A, we observed that at least four individuals were almost completely articulated and embedded by thin crusts of flowstone, which have served as protection against dissociation and bone decay (Fig. 3). Two of these individuals were collected (Fig. 2D). For others, bones are disarticulated and selectively preserved with a bias toward long bones (radius, metacarpal, etc.), tympanic bullae, and jaws. In Hoshino Cave, we entered from a natural entrance, which is not open to the public, and found an incomplete skeleton of one individual of Rhinolophus sp. several meters below a paved commercial route (Fig. 1E).

Figure 2 Various conditions of skeletal remains of Miniopterus sp.

(A) Partially articulated, not covered by flowstone; (B) complete skull with fragmentary bones, not covered by flowstone; (C) partially articulated, covered by flowstone on the cave floor; (D) almost fully articulated, covered by flowstone (NMNS-PV 23770).

Figure 3 Skeletal remains of Miniopterus. sp. on the slopes of stalagmite in Cave A.

(A) Heavily covered by stalagmite; (B) more heavily covered by stalagmite, whose locality is shown in Fig. 1D.

Fragile bones were reinforced in the lab by coating with a solution of 5% Paraloid B-72 dissolved in acetone. They are stored at the National Museum of Nature and Science, Tokyo (NMNS; Tsukuba, Ibaraki, Japan). A complete list of collected bones is provided in Table 1, and a detailed taxonomic study of the specimens is in progress.

Table 1 A list of skeletal remains of Miniopterus sp. and Rhinolophus sp. collected from Minami-Daito Island, Okinawa.

At the National Museum of Nature and Science (Tokyo, Japan), all vertebrate fossils catalogued in the Department of Geology and Paleontology start with the prefix “NMNS-PV”.

Museum ID
(NMNS-PV)	NMNS-PV branch number	Scientific name	Estimated # of individual	Identification	Cave	Field Sampling ID	
23770		Miniopterus sp.	1	Articulated body	Cave A	NA	
23771		Miniopterus sp.	1	Articulated body	Cave A	NA	
23772	1	Rhinolophus sp.	1	Left maxilla with two molars (partial)	Hoshino Cave	H160114-01-G	
2	Left mandible with teeth (m1-m3)	H160114-01-A	
3	Right mandible without teeth (partial, fragmented)	H160114-01-F	
4	Left radius (partial)	H160114-01-C	
5	Radius? (partial)	H160114-01-E	
6, 7	Pelvic bone (fragment), rib (partial),	H160114-01-B	
8	Left humerus (almost complete)	H160114-01-D	
NA	Skull (zygomatic arch only), bone fragments	H160114-01-H	
23773	1	Rhinolophus sp.	1	Left maxilla with teeth (partial)	Cave A	Y160113-10-C	
2	Left mandible (fragment)	Y160113-10-G	
3,4,5, 6,7,8,9	Right mandible with teeth (fragmented)	Y160113-10-B	
10	Left clavicle (complete)	Y160113-10-D	
11	Left radius (fragment)	Y160113-10-H	
12	Radius? (fragment, small rod branched off), bone fragments	Y160113-10-J	
13	Left humerus (partial)	Y160113-10-I	
14	Pelvic bone (almost complete), bone fragment	Y160113-10-A	
15	Metacarpal or phalange (partial)?	Y160113-10-E	
16	Metacarpal? (partial)	Y160113-10-K	
NA	Bone fragment (could not identified by YK)	Y160113-10-F	
NA	Bone fragments	Y160113-10-L	
23774	1	Miniopterus sp.	1	Left radius (fragment)	Cave A	Y160113-04-B	
2	Right radius (complete)	Y160113-04-D	
3	Left humerus (fragment)	Y160113-04-E	
4	Right humerus (complete)	Y160113-04-C	
5	Metacarpal (fragment)	Y160113-04-J	
6,7	Metacarpal or phalange? (fragments)	Y160113-04-A	
NA	Bone fragments	Y160113-04-G	
23775		Miniopterus sp.	1	Right radius (complete)	Cave A	Y160113-05-A	
23776		Miniopterus sp.	1	Right radius (almost complete)	Cave A	Y160113-06-A	
23777	1	Miniopterus sp.	1	Skull	Cave A	Y160113-08-F	
2	Left radius (partial)	Y160113-08-A	
3	Left humerus (complete)	Y160113-08-E	
4	Right humerus (fragment)	Y160113-08-D	
5	Metacarpal (fragment)	Y160113-08-G	
NA	Bone fragments	Y160113-08-B	
23778	1	Miniopterus sp.	1	Skull (partial, fragment) without teeth	Cave A	Y160113-03-A	
2,3,4,5, 6,7,8	Isolated teeth	Y160113-03-B	
9	Right dentary wo teeth	Y160113-03-K	
10,11,12, 13,14,15,16, 18,19,20,21	Left dentary, five islated teeth	Y160113-03-I	
22	Isolated tympanic bulla (partial)	Y160113-03-O	
23,24	Scapula (fragment), skull (zygomatic arch only)	Y160113-03-N	
25	Right radius (fragment)	Y160113-03-C	
26	Right humerus	Y160113-03-H	
27,28,29	Metacarpal	Y160113-03-E	
30	Metacarpal, metacarpal or phalange? (fragments)	Y160113-03-F	
31	Metacarpal (fragment)	Y160113-03-J	
32	Metacarpal or phalange? (fragment)	Y160113-03-M	
33,34,35,36	Rib	Y160113-03-D	
NA	Bone fragment (radius, left or right unidentified)	Y160113-03-G	
NA	Bone fragment (radius, left or right unidentified)	Y160113-03-L	
NA	Bone fragments	Y160113-03-P	
23779	1	Miniopterus sp.	1	Skull (maxilla with ful dentition, others fragmented)	Cave A	Y160114-01-A	
2	Right mandible with dentition (almost complete)	Y160114-01-G	
3	Isolated antemolar	Y160114-01-G	
4	Atlas (complete)	Y160114-01-E	
5	Left radius (fragment)	Y160114-01-C	
6	Right radius (almost complete)	Y160114-01-F	
7	Left humerus (complete)	Y160114-01-B	
8	Right humerus (complete)	Y160114-01-D	
9	Left femur (complete)	Y160114-01-H	
NA	Bone fragments	Y160114-01-I	
23780	1	Miniopterus sp.	1	Skull	Cave A	Y160114-02-A	
2	Left mandible	Y160114-02-D	
3,4,5	Axis, cervical bones	Y160114-02-F	
6	Thoracic vertebrate (partial)	Y160114-02-J	
7	Clavicle (complete)	Y160114-02-I	
8	Left scapula (partial)	Y160114-02-G	
9	Left radius (complete)	Y160114-02-H	
10	Left humerus (fused with other bones by calcite crystals)	Y160114-02-E	
11	Right humerus (partial)	Y160114-02-B	
12,13	Metacarpals (partial)	Y160114-02-C	
14,15,16, 17,18,19,20	Ribs	Y160114-02-K	
NA	Metacarpals or carpal (fragment)	Y160114-02-M	
NA	Bone fragment	Y160114-02-L	
23781	1	Miniopterus sp.	1	Skull (partial), skull fragments	Cave A	Y160114-03-A	
2,3,4,5,6,7	temparate bullae, isolated teeth	Y160114-03-I	
8	Left radius (complete)	Y160114-03-C	
9	Right radius (complete)	Y160114-03-B	
10	Left humerus (fragment)	Y160114-03-H	
11	Right humerus (complete)	Y160114-03-G	
12	Right femur	Y160114-03-E	
13	Metacarpal (partial)	Y160114-03-D	
14	Metacarpal (partial)	Y160114-03-J	
15	Metacarpal (partial)	Y160114-03-K	
16	Metacarpal (partial)	Y160114-03-L	
NA	Metacarpal (fragment)	Y160114-03-F	
23782	1	Miniopterus sp.	1	Left maxila with dentition (partial)	Cave A	Y160114-04-E	
2,3	Left dentary with dentition (complete), isolated m1 (trigonid basin only)	Y160114-04-C	
4,5	Right dentary with dentition (complete), isolated antemolar	Y160114-04-D	
6	Right radius (partial)	Y160114-04-B	
NA	Metacarpal or phalange (fragments)	Y160114-04-A	
NA	Bone fragments	Y160114-04-F	
23783	1	Miniopterus sp.	2	Left radius (partial)	Cave A	Y160113-07-B	
2	Left? radius (fragment)	Y160113-07-D	
3	Right radius (complete)	Y160113-07-E	
4	Right? radius (fragment)	Y160113-07-K	
5	Left humerus (complete)	Y160113-07-F	
6	Left humerus (complete)	Y160113-07-A	
7	Right humerus (complete)	Y160113-07-G	
8	Right humerus, bone fragment	Y160113-07-C	
NA	Metacarpal or phalange? (bone fragments)	Y160113-07-H	
23784		Miniopterus sp.	1	Left and right mandibles (complete, separated)	Cave A		
23785		Miniopterus sp.	1	Right maxilla with dentition (partial)	Cave A		
24999		Miniopterus sp.	1	Left mandible (partial)	Cave A		
25000		Miniopterus sp.	1	Right mandible (partial)	Cave A		

Some well-preserved fragments without secondary calcite crystal growth or acid etching were consumptively sampled for radiocarbon dating; however, none preserved collagen (i.e., no signal was detected for nitrogen using Elemental Analysis [EA] on a bone fragment), so these materials were not useful for direct dating. Because we were unable to determine the age of any individual fossil, we instead examined guano-like deposits accumulated in the same caves for time-averaged 14C ages.

Guano-like deposits on Minami-Daito Island

Guano-like deposits were sampled at four localities in Cave A (Fig. 1D) and one locality in Hoshino Cave (Fig. 1E). These deposits are accumulated in piles on the floors of the caves that can be washed downslope. They are soft muddy sediments brownish-black (10YR2/2 to 10YR2/3) to black (10YR2/1) in color. Mud particles are loosely coalesced to form sand-sized granules, which are often contaminated with dolomite fragments. They are also present in local small depressions above the floor. Flowstones in the caves are often colored black from pollution in areas where the Japanese military used them as headquarters during World War II (WWII; “Headquarter Hall” on the map) and where groundwater drains. We collected guano-like deposits from large piles (Fig. 4A) or handpicked coalesced particles using forceps (Fig. 5A). The largest pile of guano-like deposits (8∼10 m wide on the surface) was found in Hoshino Cave, which is probably the deposit noted by Shimojana (1978). The thickness of the pile is over 50 cm at a random spot where we collected samples from its bottom, a mid-part, and its top. There is no sedimentological difference observed except that the bottom sample contained more carbonate lithic fragments. In Cave A, a thin crust under a bat-lying flowstone was also sampled (Figs. 5B, 5D). The sampled deposits were dried in a convection oven at 38 °C and were sieved at 850 µm, 425 µm, and 250 µm. Samples between 250 µm and 425 µm were carefully handpicked using forceps for black particles to avoid carbonate contamination prior to further analyses.

Figure 4 Optic and SEM images of guano references and guano-like deposit in Hoshino Cave.

(A–C) Guano-like deposit (H180120g5) collected from Hoshino Cave, Minami-Daito Island; (D, E) Modern guano (R2) collected in Taito, Chiba; (F, G) Fossil guano (R3) collected in Fujido Cave, Gunma.

Figure 5 SEM and optical images of guano-like deposit in Cave A.

(A, C) Fecal pellet-like sample (Y180119g2). (B, D) Hydroxyapatite crust (Y160114g1), possibly derived from bat feces.

Because insectivorous bats consume arthropods (crickets, beetles, moths, etc.), bat guano contains their chitinous exoskeletons, which are resistant to decay. We utilized established signals of the chitinous exoskeletons of arthropods to determine whether or not the guano-like deposits are guano-derived by comparison with modern and fossil bat guanos as positive references and red soil as a negative reference. The red soil was collected from the bottom of thick soil deposits beneath a fissure in Cave A.

Modern and subfossil bat guano as positive references

For comparison with the guano-like deposits sampled in Hoshino Cave, we used a commercial guano fertilizer collected in Indonesia (called R1 in figures; ARK Co. Ltd., Ushiku, Ibaraki, Japan), modern bat guano of Miniopterus fuliginosus (R2) from an unnamed artificial cave of soft sandstone in Taito, Isumi, Chiba, subfossil guano deposit (R3) from Fujido Cave in Udeno, Gunma, and fecal pellets (R4) of Vespertilio sinensis in Chichibu, Saitama. Localities were selected by Y.K. for ease of access. These samples were used as “positive references” for comparison with our unknown guano-like samples from the Minami-Daito Island caves (see Article S1).

Scanning Election Microscopy (SEM) observation

In the fecal pellets of modern insectivorous bats, arthropod exoskeleton fragments are usually observed. Thus, we imaged dry samples of the guano-like deposits from Hoshino Cave and the four guano references at the National Museum of Nature and Science (NMNS; Tsukuba, Ibaraki, Japan) using a JSM-6510 (JEOL Ltd., Tokyo, Japan) scanning electron microscope (SEM) under 70x to 300x magnification. Images were taken at an acceleration voltage of 3 kV with gold sputter coating on the dry samples.

Fourier Transform Infrared Spectroscopy (FTIR)

Fourier transform infrared (FTIR) spectroscopy is a technique commonly used to characterize specific molecular structures in organic compounds. The FTIR spectra of chitin extracted from modern and fossil guano as well as those of bulk guano are well known based on previous studies (e.g., Wurster et al., 2010; Kaya et al., 2014), so we used FTIR to identify whether chitin is a major component of the guano-like deposits. The hand-picked samples of the guano-like deposits and reference materials (modern and fossil guano, red soil) were refluxed with 1.0M HCl at 80 °C for at least 8 h to remove exogenous carbonates and organic acids (e.g., fluvic acids) and were rinsed with deionized water until the solvent became a neutral pH. After drying in a convection oven below 38 °C, the samples were ground in an agate mortar with a pestle. They were then stored in a desiccator at room temperature until analysis.

Each powdered sample was diluted with ground KBr, pressed into a pellet in a stainless-steel disk, and analyzed without a vacuum using a JASCO FT/IR-6800 (JASCO Inc., Tokyo, Japan) at NMNS. Pure KBr was measured as the background under the same conditions as the samples. The infrared spectra of absorbance were measured from 400 cm−1 to 4,000 cm−1 by 64 scans at a resolution of 4 cm−1 and are expressed as the percentage of transmittance (%T). Automatic baseline corrections were made using a second-degree polynomial fitting (x2).

Stable isotope analyses and C:N ratio of the guano-like samples

To supplement FTIR results of the guano-like samples, we measured stable carbon and nitrogen isotope values and mass ratios of organic carbon to nitrogen (C:N) for the HCl-treated samples using a FLASH 2000 CHNS/O elemental analyzer coupled with a Finnegan MAT253 isotope ratio mass spectrometer (Thermo Fisher Scientific, Massachusetts, USA) at NMNS. Isotopic ratios are expressed in delta notation (δ13C, δ15N) in parts per thousand (‰) and reported on the VPDB scale for δ13C values and the AIR scale for δ15N values. For guano-like deposits, a thick sample from Hoshino Cave and a handpicked pellet-like sample from Cave A were selected for analysis. Three internal lab standards were analyzed for each run and used for data correction. Repeated analyses of the standards were within ±0.2‰ for δ13C and ±0.25‰ for δ15N. Carbon isotope values of guano references are used to reconstruct isotopic signals of their diet by taking the following enrichment factors into consideration. The enrichment factor between the feces of insectivorous bats and their diet (i.e., insects) is known to be negligible (Salvarina et al., 2013), whereas larger enrichment factors are observed between the cuticles of insects and their diet of plant matter. In an experimental study, Gratton & Forbes (2006) showed that tissues of predacious beetles are more enriched by 2.2‰ relative to their aphid diet.

Stable carbon isotopes of bone apatite

Stable carbon isotope analysis of animal tissues is a useful technique to detect an isotopic signal of C4 plants in their habitat. Although no C4 plant is native to the Daito Islands, historical records (Editorial Committee of the History of Minami-Daito Village, 1990) document that the C4 perennial grass sugarcane was introduced on Minami-Daito Island in 1899–1900 and increasingly replaced native C3 forests on the island by the 1930s (see Introduction). We therefore hypothesize that any organic materials on the island with the isotopic signature of C4 plants must have originated after the introduction of sugarcane in 1899–1900. Thus, stable carbon isotope analysis with correct applications of fractionation factors can serve as a time constraint for direct human influences on Minami-Daito Island. Note that no scientific study such as palynological analysis of sediment cores is available to independently validate the historical documents.

Relatively well-preserved bat bones (no acid etching, no or limited calcite crystallization) from Cave A were selected for carbon isotope analysis of bone apatite. For each bone, the surface was shaved off to remove calcite crystals and carbonate deposits as much as possible from the thin bones. The powdered samples were soaked in 2% NaOCl overnight for at least 12 h at the room temperature. After rinsing with deionized water, they were treated with buffered 0.1M acetic acid (pH = 4.2) for 2 h at 40 °C until no obvious bubbles were observed, and they were neutralized with ultrapure water and dried in an oven at 40 °C. The treated samples were sent to Shoko Science Co., Ltd. (Saitama, Japan) and analyzed for stable carbon isotope via a H3PO4 digestion using by a continuous flow Thermo Scientific Delta V Plus isotope ratio mass spectrometer coupled to a Thermo Scientific GasBench II. Of the eight samples analyzed in this study, seven come from individual bones, whereas one sample is composed of multiple small bone fragments.

The carbon isotope values of bone apatite were used to reconstruct isotopic signals of vegetation by applying experimentally known enrichment factors (ɛ*) and binary mixing between C3- and C4-endmembers. The enrichment factor applied in this study is +11.0 ± 0.1‰ between bone apatite and insectivorous diet (Podlesak et al., 2008). For isotopic spacing (δ) between insectivorous diets of bats and surrounding vegetation, we applied +0.8‰ or +3.0‰ based on an a dietary-switch experimental study by Gratton & Forbes (2006). For the end-members in binary mixing, δ13C values of −12.5‰ and −29. 1‰ were used for pure C4 (Cerling et al., 1997) and C3 vegetation, respectively. The latter was calculated based on the equation of Kohn (2010), which takes mean annual precipitation (MAP), altitude, and latitude into account. MAP of Minami-Daito Island between 1942 and 2020 was calculated as 1680 mm/year, which was applied to the equation. See Article S1 for more details.

Radiocarbon dating of subfossil guano references

We determined the 14C age of both the fossil guano reference (R3) and the guano-like samples for comparison. The same samples analyzed for stable isotopes were chosen for radiocarbon dating. Prepared samples were sent to the accelerator mass spectrometry (AMS) facility at the University Museum, the University of Tokyo, where the samples were further prepared and measured by Compact AMS System (National Electrostatics Corp., Wisconsin, USA), following their in-house protocols (Omori et al., 2017). For each sample, δ13C measured by AMS was used to correct for the conventional radiocarbon age (Stuiver & Polach, 1977). These radiocarbon ages were calibrated by IntCal20 curves (Reimer et al., 2020) in OxCAL4.2 (Bronk Ramsey, 2009) and are expressed as calBP (Before Present; 0 calBP = AD 1950). Preparation and analytical methods are described in Article S1.

Results and Discussion

Skeletal remains of locally extinct cave-dwelling bats

In a survey during the 1970s, Shimojana (1978) reported a few skeletons of Rhinolophus sp. from Hoshino Cave. In this study, we found the skeletons of single individuals of Rhinolophus sp. in both Hoshino Cave and Cave A. Additionally, we collected skeletons of Miniopterus sp. from Cave A, which represent at least 13 individuals (taxonomic study in progress). Some skeletons are articulated, but many are disconnected with associated bones closely scattered on the surface of the cave floor (Fig. 2), so we infer that they were not transported by water after death. No traces of soft tissues such as pelage are preserved. In Cave A, we observed at least four individuals of Miniopterus sp. that were embedded within a thin crust of flowstone (Fig. 3), two of which were collected. No fossil remains are permineralized, but many are encrusted with calcite. The condition of calcite crystal growth on and in bones depends on the availability of mineral-rich water flow near the bones. In many bone materials, the surface is partially covered with small calcite crystals, and/or the medullary cavity of long bones is partially infilled.

Due to the lack of bone collagen for radiocarbon dating in the specimens we sampled, it is unknown whether these bat species utilized the cave contemporarily; however, because there is no difference in bone preservation and secondary crystal growth of calcite between the two species, our paleontological interpretation is that the ages of their existences do not differ even if they may not be coeval sensu stricto.

SEM observation of guano-like samples

Fragments of insect exoskeletons are clearly visible in both modern (Figs. 4D–4E) and fossil guano references (Fig. 4G) at or above 70x magnification although the fossil guano is degraded to be humus-like and contaminated with sediments (Fig. 4F). On the other hand, the guano-like samples appear to be aggregates of clay-sized particles with no clear presence of insect remains (Figs. 4B–4C), suggesting that they may not be composed of bat guano. Nevertheless, because the preservation potential of chitinous insect remains is expected to be lower in guano deposits from tropical wet caves like those on Minami-Daito Island than in the fossil guano reference collected from a more temperate region, we proceeded with further analyses.

FTIR

None of the guano-like deposits collected in this study present spectral patterns characteristic of fresh or fossil guano (Figs. 6 and 7, see Article S1 for full description of references). In the range between 1,000 cm−1 and 1,800 cm−1, different spectral patterns are observed between guano references and sampled deposits. Particularly, a shoulder peak at 1,712 cm−1 associated with C=O stretching of carboxylic acid (Dick, Santos & Ferranti, 2003) is present in guano-like samples but does not occur in the spectra of guano references (Fig. 6). This peak occurs in humified organic matter (Dick, Santos & Ferranti, 2003; Palumbo et al., 2018).

Figure 6 FTIR spectra of guano references, guano-like deposits, and humic acid.

(A) Commercial fertilizer (R1); (B) Modern fecal pellet (R3); (C) Fossil guano (R4); (D, E) Guano-like deposits sampled in Cave A; (F, G) Guano-like deposits sampled in Hoshino Cave; (H) humic acid (Dick, Santos & Ferranti, 2003); (I) humic acid extracted from municipal solid waste (Palumbo et al., 2018). The numbers of Minami-Daito samples (4: Y180119g3, 5: Y180120g1, 6: H180120g5) correspond to sampling localities in Figs. 1D–1E. u: upper, b: bottom.

Figure 7 FTIR spectra of lateric soil, fissure-filled deposit, sampled deposits, and hydroxyapatite.

(A) Whole soil of Latosol (Dick, Santos & Ferranti, 2003); (B) Fissure-filled deposit; (C) Guano-like deposit; (D) Hydroxyapatite crust; (E) Hydroxyapatite (Jayaweera et al., 2018). The numbers of Minami-Daito samples (1: Fissure-filled deposit, 2: Y160114g1, 3: Y180119g2) correspond to sampling localities in Fig. 1D.

The FTIR spectrum of the guano-like sample Y180119g3 (Fig. 6D) is comparable to that of humic acid (Figs. 6H–6I). The absorption at 1,078 cm−1 is ascribed to the stretching vibration of oxygen bonds in aliphatic ether (Zhang et al., 2015). The presence of aliphatic compounds is consistent with the interpretation that this sample is a fraction of humic substances (Méndez, 1967). The peak at 1,244 cm−1 can be interpreted as antisymmetric C-O stretching and OH deformation in a carboxyl group, and -C-OH bending of phenols and tertiary alcohol in humic substances (Dick, Santos & Ferranti, 2003). As organic functional groups, carboxylic acids are contained in humic substances. We interpret that the guano-like deposits sampled in this study are forms of humic substances rather than fossil bat guano, with the exception of Y160114g1, which was identified as hydroxyapatite (Fig. 7D) by comparing characteristic spectral peaks with those of pure hydroxyapatite in Jayaweera et al. (2018).

The hydroxyapatite sample, Y160114g1, occurs as a thin crust light brown in color only partially exposed under a fracture of more recent flowstone on the slope of a large stalagmite (Fig. 5B). Interestingly, we found skeletal remains of at least two individual bats lying on a slope of the stalagmite, both of which are also covered by recent flowstone (Fig. 3B). The hydroxyapatite crust stratigraphically underlies the bat skeleton. As a calcium phosphate mineral, hydroxyapatite is commonly precipitated in caves where phosphorous-rich organic deposits such as bat guano or bones are supplied and interaction between the guano-derived leachate and calcium-rich water from carbonate host-rock can occur at a pH greater than 6 (Fiore & Laviano, 1991; Hill & Forti, 1997; Tămaş, Miheţ & Giurgiu, 2014; Giurgiu & Tămaş, 2013). Because this hydroxyapatite crust was found in association with a bat skeleton, and hydroxyapatite precipitation is inhibited in the presence of humic and fluvic acids (Inskeep & Silvertooth, 1998), this crust was likely derived from bat guano rather than humified plant material.

C:N ratios, δ13C, and δ15N values of the guano-like samples

The C:N ratios, total nitrogen by weight percent (%N), δ13C values, and δ15N values of the guano-like samples are shown in Fig. 8 (Table 2, Table S1). In the bulk sample of the fissure-filled deposit, total nitrogen is nearly 0.1%, which is within the standard range of nitrogen content for sediments (Forbes & Bestland, 2006). The nitrogen content in the fecal pellet-like sample Y180119g2 (Fig. 5A) from Cave A is lower than the %N range of guano references (Fig. 8A). Because the FTIR spectrum of the pellet-like sample similarly traces that of the fissure-filled deposit, we interpret that the fecal pellet-like sample is a mix of sediment and organic matter. In general, C:N ratios of all analyzed samples are tightly clustered, ranging from 4.2 for fissure-filled deposit to 6.5 for the fossil guano reference with the unknown guano-like samples (4.8 on average) equivalent to those of modern guano (R1 and R2) (Fig. 8B). These values are much lower than fresh plants and coals (e.g., Schmidt & Gleixner, 2005) and superficially meet one of Forbes & Bestland’s (2006) criteria to identify fossil guano (C:N < 10). Nevertheless, in our case, the low C:N values of the samples are still consistent with the FTIR-based classification of the red soil of the Minami-Daito Islands as a lateritic soil because it is known that C:N ratios in humic substances and soil organic matter extracted from modern lateritic soil show roughly comparable values between 7 and 10 (Dick, Santos & Ferranti, 2003).

Figure 8 Isotopic composition of carbon and nitrogen, total nitrogen by weight percent (%N), and C:N ratio in the selected guano-like samples and guano references, corresponding to Table 2.

(A) Stable nitrogen isotope values vs. %N; (B) stable carbon isotope values vs. C:N. u: upper, m: middle, b: bottom. Raw data are provided in Table S1.

Table 2 Summary of carbon and nitrogen isotopes, C:N ratios, weight percent of carbon and nitrogen in samples.

Sample ID	Material type	Sampling locality	N	δC (‰, VPDB)	δN (‰, AIR)	C/N	%C in sample	%N in sample	
				Average	SD	Average	SD	Average	SD	Average	SD	Average	SD	
R1	Modern guano (R1)	Commercial fertilizer	5	−27.1	0.22	6.5	0.37	5.2	0.24	41.8	1.0	8.1	0.5	
R2	Modern guano (R2)	Taito, Chiba	1	−28.2		5.9		5.0		44.8		8.9		
R3	Modern fecal pellet (R3)	Chichibu, Saitama	3	−19.1	0.03	3.0	0.10	6.0	0.02	51.1	0.4	8.6	0.1	
R4	Fossil guano (R4)	Fujido Cave, Ueno, Gunma	1	−25.6		11.6		6.5		53.8		8.3		
Sediment	Fissure-filled cave deposit	#1 in Cave A	7	−23.9	0.15	9.5	0.17	4.2	0.24	0.5	0.03	0.1	0.002	
Y180119g2	Fecal pellet-like sample	#3 in Cave A	1	−26.1		10.2		4.8		13.2		2.7		
H180120g5-u	Guano-like deposit	#6 in Hoshino Cave	1	−26.4		6.1		4.8		44.3		9.2		
H180120g5-m	Guano-like deposit	#6 in Hoshino Cave	1	−26.7		5.8		4.6		40.9		8.8		
H180120g5-b	Guano-like deposit	#6 in Hoshino Cave	1	−26.5		5.9		4.8		43.2		9.1		

The δ13C values of guano references range from −28.2‰ to −19.1‰, encasing the guano-like samples (mean ± 1SD = −26.4 ± 0.22‰) (Fig. 8B). Accounting for the isotopic spacing among bat guano, their insectivorous diet, and surrounding vegetation, the guano-like deposits are well within the range of C3 plants, which exhibit a general range from −23‰ to −37‰ in non-desert areas (Kohn, 2010). As they completely lack a signal of C4 plants, we suggest that these deposits were formed before anthropogenic influences because sugar cane, C4 perennial grass, became exclusively dominant by the 1930s due to heavy cultivation on the island.

The δ15N values of the referenced guano samples widely range from 3.0‰ to 11.6‰, placing the guano-like deposits from Hoshino Cave in its mid-range and the fecal pellet-like sample from Cave A close to its upper limit (Fig. 8A). Forbes & Bestland, (2006) used high δ15N values above 12‰ in association with low C:N ratios below 10 and high contents of SO3+P2O5+CaO to identify a potential guano layer in accumulated cave deposits. The δ15N values of the guano-like samples from Hoshino Cave do not meet their criteria although neither do those of some guano references. Overall, as in the case of %N and C:N ratios, stable carbon and nitrogen isotopes were not good indices to identify bat guano in caves of tropical islands where lateritic soil can be developed.

Stable carbon isotopes of bone apatite

Bone apatite δ13C values (n = 8) show a wide range from −15.0‰ to −8.36‰ with a mean of −11.5 ± 2.38‰ (±1SD) (Table 3). Fig. 9A shows estimated isotopic signals of vegetation, calculated from carbon isotope values of bone apatite of bats. Considering that enrichment factors and isotope values in the mixing model are reasonably assumed (see Article S1), C4-feeding insects explains more than 10% of the total diet in six out of the eight individuals analyzed in this study even with a large isotopic spacing between predatory insects and plants (open circle in Fig. 9A). On average, the estimated δ13C value of bulk plants is −25.4‰ (open circle), which is translated to 22% of C4 plants in the simple mixing model. On Minami-Daito Island, C4 plants were introduced only by humans via sugarcane plantations. Therefore, the unmistakable isotopic signal of C4 plants demonstrates that these individuals coexisted with humans after 1900.

Table 3 Stable carbon isotope values of bone apatite in the Minami-Daito cave-dwelling bats.

Individual #	Element	δ13C (‰, VPDB)	Lab Code	
1	Proximal end of right humerus	−15.0	Y160113b-6	
2	Multiple bone fragments	−14.0	Y160113b-201902-4-2	
3	Distal end of right humerus	−12.4	Y160113b-8	
4	Left radius	−12.3	Y160113b-201902-1	
5	Right radius	−11.3	Y160113b-201902-2-2	
6	Distal end of humerus	−9.6	Y160113b-5	
7	Proximal end of femur	−9.1	Y160113b-7	
8	Fragments of left humerus and left radius	−8.3	Y160113b-201902-3-2	

Figure 9 Estimated δ13C values of vegetation based on bone apatite of cave-dwelling bats and chronological ranges of the extirpated bats on Minami-Daito Island.

(A) Estimated δ13C values of vegetation on Minami-Daito Island, calculated from bone apatite δ13C values of the cave-dwelling bats. Measured δ13C values of bone apatite are provided in Table 3. For the isotopic enrichment factors and δ13C values of pure C3- and C4-vegetation applied in this study, see Supplementary Article S1 for details. (B) Estimated ages of the extirpated Minami-Daito bats with lines of evidence revealed by this study. Thicker and darker lines indicate the time range that is more strongly supported.

Group size and estimated ages of cave-dwelling bats on the Daito Islands

Two locally extinct bat species, Miniopterus sp. and Rhinolophus sp., once existed in the lifted atoll remotely located from the closest landmass. Based on the number of skeletal remains recovered from Cave A in this study (at least n = 17 for Miniopterus sp. and n = 1 for Rhinolophus sp.), this cave was utilized by more individuals of Miniopterus sp. than Rhinolophus sp. Considering the taphonomic conditions of the Minami-Daito bats (unpermineralized, exposed on cave floor without any depositional cover, fast crystal growth around bones in tropical wet caves), we interpret that these skeletons are only a few thousand years old at most but probably even younger and that the two species occurred contemporarily in the paleontological sense.

FTIR spectra and SEM images suggest that the guano-like deposits on Minami-Daito Island are most reasonably humic substances, which are the final constituent of the physicochemical degradation and microbial decomposition of organic matter, particularly plant materials (Stevenson, 1994). The hydroxyapatite crust under the bat skeleton was possibly formed from the interaction between bat guano and limestone/dolomite. Nevertheless, we reject the null hypothesis that the “guano deposits” observed by Shimojana (1978) are fossil bat guano and deny the presence of large amounts of guano deposits in the caves, which suggests the skeletal remains are not resultant from large populations or those which utilized these caves for generation after generation.

Our humic samples from both Hoshino Cave and Cave A yielded similar ages of 4,565 calBP (on the 3-point average) and 4,640 calBP, respectively (Table S2), which we think provide the maximum age constraint for the bat species as the bat bones collected in this study were scattered nearby pellet-like humic particles in Cave A. A better age constraint was provided by stable carbon isotopes of the selected bat bones from Cave A. Isotopic signals of C4 grass-feeding insects detected in various degrees from six out of eight individuals strongly indicate that some of the sampled bats lived after 1900 (Fig. 9B).

In summary, small groups of bats that were dispersed to Minami-Daito Island did not survive long due to strong anthropogenic stresses on the islands. Whether or not their arrivals were caused by a single sporadic event or multiple events is still inconclusive in this study. However, future taxonomic studies and analyses of morphological variation could elucidate the possible biogeographic origins of the strayed populations, which would provide a hint to understand how these species ended up co-occurring on the small island.

Extirpation of two cave-dwelling bat species on the Daito Islands

Our multiproxy approach to constrain the estimated ages of the Minami-Daito cave-dwelling bats include hearing surveys of local residents on the Daito Islands. This survey was conducted by Dr. Hidetoshi Ohta on August 22 and October 28 in 1991 for a different purpose but happened to provide an imperfect yet interesting perspective regarding the extirpation of the bats (pers. comm. with H. Ohta on December 31, 2018 by YK and DF). According to his field notes, three men (Mr. Akamine, Mr. Nakama, Mr. Sunagawa; all in their 80s) commented that they witnessed “bats smaller than the Daito flying fox” in a cave during the daytime within a few years after the end of WWII. In the second survey, three other men (Mr. Toma, Mr. Suzuki, and Mr. Okiyama; two in their 80s, one in his 90s) commented that they also saw “baby flying foxes” in caves near “Hagu (rim)” before WWII. For local residents, the Daito flying fox is a typical mammal on the island. Thus, these bats are reasonably considered as Miniopterus sp. and/or Rhinolophus sp., as the Daito flying fox roosts in the canopies of trees during the daytime, not utilizing caves. If our interpretation is correct, a remnant population of these bats persisted until sometime after WWII, and the very last survivors could have been present until the 1950s, which is consistent with our taphonomic interpretation of the bones.

On Minami-Daito Island, the irreversible rapid change from tropical/subtropical forests to sugarcane fields was extraordinary following the arrival of early settlers/residents in 1899-1900 with heavy development by the 1930s. In addition to reducing favorable habitats for cave-dwelling bats by opening the land and filling in sinkholes, there would have been other mortality causes. Many caves were used as natural storage and air-raid shelters during WWII. Cave A was a base of the Imperial Japanese Army during WWII. There are metal scraps, glass bottles, and wood plausibly derived from that age. Surface water also transports waste into caves. We found coal-like carbon particles in the guano-like samples while handpicking in the lab, which may be related to wasted carbon derived from old steam locomotives that existed from 1917 to 1983 to transport harvested sugarcane stalks as the railway track was run near Cave A (Kada, 2009). Chemical pesticides were introduced in sugarcane plantations, and the aerial spray of pesticides was conducted at least in the early 1970s (Editorial Committee of the History of Minami-daito Village, 1990: p.1167). Considering the extremely rapid development on the small island, anthropogenic activities are undeniably responsible for the local extinctions of the last population of the cave-dwelling bat species. It is noted however that the results of our study do not completely exclude the possibility that these bats were non-viable drifting populations that naturally died out without leaving descendants. In the future, this could be tested by finding out how (and whether) other caves on the Daito Islands were utilized by the extirpated bats.

Conservation paleobiology on tropical oceanic islands

Using a multidisciplinary approach, we were able to bracket the estimated chronological age of the Minami-Daito cave-dwelling bats, ranging from a maximum constrained age of 4,640 calBP (radiocarbon age of humin closely scattered with the bat bones in Cave A) to the 1950s (hearing survey). Stable carbon isotopes of bone apatite provided unmistakable direct evidence that they existed after 1900 and reinforced the hearing survey record.

Despite rapid civilization and deforestation on the island throughout the 20th century, the Daito flying fox has been able to survive on the island. The Daito flying fox, which is designated as a critically endangered subspecies (IA) in the National Red Data List of the Ministry of the Environment, Japan (Japanese Red List, 2020) and is thus of high concern for species conservation (Vincenot, Collazo & Russo, 2017), consumes fruits and nectar of native species such as Chinese banyan (Ficus microcarpa), Fiscus superba var. japonica, Ficus virgala, and fan palm trees (Livistoma chinensis var. amanoi), but it also utilizes windbreak trees and street trees as a food source (Kinjo, 2009). Flexible food preferences may explain how the Daito flying fox has survived throughout anthropogenic damages as in the case of the neighboring subspecies Orii’s flying fox (Pteropus dasymallus inopinatus) on Okinawa Island, which utilizes planted trees in urban areas as stable food sources (Nakamoto, Kinjo & Izawa, 2007).

Regarding the presence of fossil bat species on an oceanic island, a similar case was reported for an extinct endemic bat, Synemporion keana (lava-tube bat), from the Hawaiian Islands, which are over 3800 km away from the nearest continental landmass (Ziegler, Howarth & Simmons, 2016). Synemporion keana is a vespertilionid bat slightly smaller than the extant Hawaiian hoary bat (Lasiurus cinereus semotus) that is known to have inhabited the Hawaiian Islands at least until the earliest Polynesian cultural period because a fossil of S. keana was successfully radiocarbon dated to 1,670 BP (Ziegler, Howarth & Simmons, 2016). According to that study and the references therein, those fossils were recovered from infilled deposits (soil, eolian deposit, pond deposit) or exposed on the floor of limestone sinkholes, volcanic lava tubes, a piping cave, and a volcanic-tuff cone crater. Similar to our studies, those fossils were not suitable for DNA extraction or radiocarbon dating. They estimated a chronological range of the Hawaiian fossil bat species by establishing the 14C ages of both coexisting animal skeletons such as birds and rats and cultural artifacts derived from the same stratigraphic level or nearby and the formation of volcanic caves from which the fossils were found. Another interesting case of the late Holocene (<4,000 BP) extirpation of bats is found on a tropical island in the Caribbean (Soto-Centeno & Steadman, 2015).

In this study, we revealed that two cave-dwelling bat species were extirpated from Minami-Daito Island without any zoological record when they existed. Minami-Daito Island used to be a home of three bat species, but only a single species survived anthropogenic disturbance. Geological, geochemical, and paleontological approaches are useful to bracket chronological ranges of the two extirpated species. Chronological ranges of extirpated species should be estimated with higher accuracy to deduce causality of the local extinction events and subsequently to promote conservation of biodiversity on oceanic islands, which are generally vulnerable to anthropogenic factors.

Conclusions

Minami-Daito Island is an oceanic island that has never been connected to another landmass since its emergence. A few skeletons of Rhinolophus sp. were found during the 1970s from Hoshino Cave, the only tourist cave on the island (Shimojana, 1978). In this study, we found not only an additional skeleton of Rhinolophus sp. from Hoshino Cave but also a skeleton of the same species and more of Miniopterus sp. from another private cave. Because these bones, which were exposed on the floor of the tropical dolomite caves, did not preserve collagen for radiocarbon dating, we used multidisciplinary proxies to provide time constraints for estimated ages of the bats. Differing from a previous study, we obtained limited witnesses of cave-dwelling bats from before/during WWII and concluded the “guano-like” deposits previously identified were humic substances formed ca. 4,640 calBP, which gives the maximum age constraint for the extirpated bats. Their populations were never large and quickly shrank due to rapid and heavy deforestation for sugarcane plantations on the island. They were extirpated from Minami-Daito Island without any zoological record when they existed. This study highlights that anthropogenic activities increase the mortality of animal species newly strayed into insular environments.

Supplemental Information

Supplemental Information 1 Isotopic composition of carbon and nitrogen, total nitrogen by weight percent (%N), and C:N ratio in selected guano-like samples and guano references, corresponding to Table 2 and Fig. 8

Click here for additional data file.

Supplemental Information 2 Summary of radiocarbon ages (calBP) of a fossil guano reference and Minami-Daito samples

Click here for additional data file.

We deeply appreciate Shin-Ichiro Kawada (National Museum of Nature and Science, Japan for permitting us to access zoological collections at NMNS and assisting with fieldwork and Benjamin T. Breeden III (University of Utah) for providing constructive comments and English proofreading for the early draft. We are thankful to Katsuji Yoshida (Cave Exploration Pro-Guide Team Ciao!), Takeshi Matsushita (Cave Exploration Pro-Guide Team Ciao!), and Yukari Yamaguchi (Cave Exploration Pro-Guide Team Ciao!) for making a route map of Cave A, to Ken O ¯yabu and Masaharu Hayakawa (Uekusa Gakuen University & Junior College) for permitting access to modern bat guano in Taito, Isumi, Chiba, to Minoru Yoneda (University of Tokyo) and Takayuki Oomori (University of Tokyo) for AMS radiocarbon dating, to Ritsuro Miyawaki (NMNS) for FTIR, and to Hidetoshi Ohta (Museum of Nature and Human Activities, Hyogo) for providing his field notes. We are grateful to Yushi Osawa, Keiko Osawa, Yu Iijima, and Hiroko Nagaoka for field assistance and to Mika Yagishita and Sonoko Suzuki for lab assistance. Nozomi Suzuki provided us great support for isotope analyses. We thank Brandon Hedrick for carefully handling our manuscript and J. Angel Soto-Centeno and one anonymous reviewer for constructive comments on the early draft, which improved the quality of the manuscript.

Additional Information and Declarations

Competing Interests

Author Contributions

Field Study Permissions

Data Availability

Kazuaki Higashi is an employee of the Office Key Point, Japan.

Yuri Kimura conceived and designed the experiments, performed the experiments, analyzed the data, prepared figures and/or tables, authored or reviewed drafts of the paper, and approved the final draft.

Dai Fukui, Mizuko Yoshiyuki and Kazuaki Higashi performed the experiments, authored or reviewed drafts of the paper, and approved the final draft.

The following information was supplied relating to field study approvals (i.e., approving body and any reference numbers):

This fieldwork was conducted outside national parks or restricted areas designated by the Nature Conservation Act. Prior to the field study, we fully confirmed that this particular fieldwork did not require an official permit from any level of governmental managements.

The following information was supplied regarding data availability:

The complete list of fossils is available in Table 1. The raw measurements of stable isotope composition of carbon and nitrogen are available in Table S1.

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
