# Peer review of "Conservation paleobiology on Minami-Daito Island, Okinawa, Japan: anthropogenic extinction of cave-dwelling bats on a tropical oceanic island"

_PeerJ, doi:10.7717/peerj.12702_

## Round 0.1 · original submission · Minor Revisions

Dear authors,

Thank you for your submission to PeerJ. Following suggestions from reviewers and a careful reading of the manuscript, I believe this paper will be suitable for publication in PeerJ following minor revisions. I found this paper to be quite interesting and am excited to see it published.

Both reviewers noted that the paper lacked some focus. I agree strongly with reviewer 1's idea that the hypotheses need to be laid out more clearly at the end of the introduction (with the addition of a hypothesis regarding what led to the extinction of the bats on the island). I would also suggest you amend the abstract to clarify the hypotheses. I wasn't sure what your goals were after only reading the abstract.

When you submit your revisions, please include a tracked changes version of the manuscript, a clean version of the manuscript, as well as an itemized annotated reviewer response document.

Please let me know if you have any questions.

Best,

Brandon P. Hedrick, Ph.D.



In addition to reviewer comments, I suggest the following changes:

How many miles from Okinawa is Minami-Daito? I am not sure what the range is for these bat species, but how can you rule out multiple colonization events? Could you discuss this a bit in the discussion? Reviewer 2 suggests something along similar lines.

Line 78: Perhaps ‘natural testing grounds’.

Line 82: ‘against further anthropogenic extinction’

Line 83: Do you have a reference you could add for this statement?

Line 84: ‘remote islands’

Line 90–91: Perhaps ‘rapid development of sugarcane cultivation along…’

Line 139: ‘Lower Daito Layer’ spelling

Line 155: Either ‘There is a countless number of sinkholes’ or ‘there are countless sinkholes’

Line 168: ‘suggesting these individuals were not transported by water flow’.

Line 195–196: There seems to be a grammatical issue with this sentence and I’m not sure what you mean. Please revise.

Line 304: ‘was 1680 mm/year’

Line 332: ‘by water after’

Line 444: How many skeletal remains were found? Can you add ‘n = ??’ here to clarify?

Line 465: Grammar. What do you mean by ‘might have been strayed’?

Line 498: I think ‘development’ is probably a better word than ‘civilization’ here. Also line 508.

Line 536: ‘survived anthropogenic disturbance’

Please make the scale bars in figure 5 (particularly 5C) larger. They require a lot of zooming in on the image to see the scale.

·

Basic reporting

This manuscript focuses on documenting the extirpation (i.e. population level loss) of two species of bat in the genera Miniopterus and Rhinolophus from Minami-Daito Island. These extirpations are interesting from the standpoint of recent changes in the bat community composition of the island, where another bat, Pteropus dasymallus, is now the only extant mammal. The aim of the study is listed; however, given that it is presented in light of previous observations, I feel like this aim could have been better presented as a clear hypothesis. Similarly, the aim only focused on the test of guano-like deposits, yet I feel like there is another hypothesis that was overlooked in the presentation of the introduction that relates to the cause of extinction of these bats. Improving the clarity of how these hypotheses are presented could really help. Overall, the ideas in manuscript are well organized and for the most part the text is written clearly – although there are various suggestions where the writing could be improved for clarity and tidiness of grammar (see attached PDF with comments).

Experimental design

The experimental design used is appropriate to document these bat losses. The multiple evidence approach used combining isotopic, SEM, and radiocarbon data, and the link to local historical accounts of the presence of these bats was well stitched in, and it does provide potential evidence for the influence of anthropogenic change on the bat community of Minami-Daito Island. Refining/clarifying the hypotheses tested would really make the manuscript flow more logically. The methods used were presented clearly and well described.

Validity of the findings

The authors presented a multi-prong approach to document the loss of the bat genera Miniopterus and Rhinolophus from Minami-Daito Island. The evidence presented does support the finding and conclusions that these bats were recently extirpated. However, while the timing for the loss of these bats presented does match with fast anthropogenic habitat transformation, the study does lean a lot on that correlation and not on direct evidence. For example, given the low number of individuals found (i.e. MNI = 17 Miniopterus and 2 Rhinolophus), it could be also possible that these were non-viable drifting populations that naturally died. The direction of the conclusions by the authors is not incorrect, but because there could be other plausible explanations for the extirpation of these bats, perhaps the conclusion should be toned down and balanced by adding alternative explanations.

Additional comments

See line comments in the attached PDF.

Reviewer 2 ·

Basic reporting

The abstract could use a little rewording, e.g., the first sentence is a little unclear due to word choice of filtration multiple times "With strong environmental and geographic filtration, vertebrates incapable of
flying and swimming are strongly filtered out of island ecosystems." and immediately noting that "the skeleton of an extinct cave-dwelling bat and fossil guano were briefly reported in a previous study" introduces confusion early on and takes away from the potential significance of the new research. I continued to find reference to this throughout the paper a bit distracting, particularly in the discussion when the authors note that taxonomic revision is ongoing.

Experimental design

I appreciate the multi proxy nature of this research and how the authors brought together multiple lines of data to tell a story. I wonder if there is a way to independently validate the historical claims that C4 appeared in about 1900, such as looking at a contemporaneous pollen core, historical photographs, or herbarium specimens? Is there any possibility of a marine influence in the diets of these bats, and if so, could you use MarineCal rather than Intcal?

Validity of the findings

The findings appear valid, though as I mention there are methodological caveats for the authors to consider and elaborate on regarding apatite.

Additional comments

This is a really exciting and well done study. I enjoyed reading the paper and look forward to learning more about this system from the authors! While it is unfortunate that the bones did not yield collagen - which would have been the most accurate choice for radiocarbon dating - I think the authors did the best they could to find other potential constraints on the age of the specimen, and they do a good job acknowledging this methodological uncertainty and downside.

I really like how the authors connected their data to a conservation perspective.

I think a way to improve this manuscript is to incorporate a bit more discussion about the downsides and constraints of apatite in isotopes and be more explicit about their fractionation estimates.

---

## Round 0.2 · accepted · Accept

Dear authors,

Thank you for your careful attention to reviewer comments. I now find this paper to be publishable in PeerJ and am moving it to the next stage. However, on a final reading I noted a number of grammar issues that should be rectified before publication.

Line 70: What do you mean by ‘did not leave many generations’? Reword?

Line 73–74: ‘hearing surveys to local residents’ has grammar issues. Perhaps ‘of local residents’?

Line 95: Should be a – between 1899–1900 rather than a hyphen

Line 320: accidental space in -29.1

Line 359: Change ‘contemporarily’ to ‘at the same time’

Line 418: “for a fissure filled…’

Line 451: Accidental space between ‘10%’

Line 510: Should be a – between 1899–1900 rather than a hyphen

Line 585–586: ‘newly strayed’ is odd wording. Perhaps if this doesn’t change your meaning: ‘increase the mortality of animal species that recently colonize insular environments’

Please make sure to take care of them prior to the proof process.

Thanks so much for your submission to PeerJ. I found this paper to be very interesting! Please let me know if you have any questions.

Best,

Brandon P. Hedrick, Ph.D.